# Long-Term Elite Controllers of HIV-1 Infection Exhibit a Deep Perturbation of Monocyte Homeostasis

**DOI:** 10.3390/ijms26093926

**Published:** 2025-04-22

**Authors:** José M. Benito, Daniel Jiménez-Carretero, Jaime Valentín-Quiroga, Ignacio Mahillo, José M. Ligos, Clara Restrepo, Alfonso Cabello, Eduardo López-Collazo, Fátima Sánchez-Cabo, Miguel Górgolas, Norma Rallón

**Affiliations:** 1HIV and Viral Hepatitis Research Laboratory, Instituto de Investigación Sanitaria Fundación Jiménez Díaz, Universidad Autónoma de Madrid (IIS-FJD, UAM), 28003 Madrid, Spain; 2Hospital Universitario Rey Juan Carlos, 28933 Móstoles, Spain; 3Unidad de Bioinformática, Centro Nacional de Investigaciones Cardiovasculares (CNIC), 28029 Madrid, Spain; 4Grupo de Respuesta Inmune Innata, IdiPAZ, Hospital Universitario La Paz, 28046 Madrid, Spain; 5Department of Statistics, Instituto de Investigación Sanitaria Fundación Jiménez Díaz, Universidad Autónoma de Madrid (IIS-FJD, UAM), 28049 Madrid, Spain; 6Cytek Biosciences, Inc., Fremont, CA 94538, USA; 7Hospital Universitario Fundación Jiménez Díaz, 28040 Madrid, Spain

**Keywords:** HIV infection, elite controllers, monocytes, immune phenotype, clustering analysis

## Abstract

Elite controllers (ECs) represent a unique subset of people living with HIV (PLWHs), who can suppress viral replication without requiring antiretroviral therapy (ART). However, despite this viral control, ECs exhibit increased incidences of various comorbid conditions and heightened systemic inflammation, which has been linked to monocyte activation. In this study, we performed an in-depth phenotypic analysis of monocytes in a cohort of long-term ECs (LTECs) and compared them to non-controller patients with ART-mediated control of HIV replication and to non-controller patients with uncontrolled viral replication. A total of 67 participants were included: 22 LTECs, 15 non-controllers on ART (onART), 10 non-controllers without ART (offART), and 20 uninfected controls (UCs) as a reference group. Monocyte phenotypes were analyzed using spectral flow cytometry with a 13-marker panel. The data were analyzed using two approaches: (a) FCS Express software v.7 to define different subsets of monocytes and assess the levels of expression of eight different monocyte functional markers and (b) R software v.4.1.1 for unsupervised multidimensional analysis, including batch correction, dimensionality reduction, and clustering analysis. Monocyte phenotypic profiling was conducted using three different approaches: (1) assessment of monocyte subsets (classical, intermediate, and non-classical monocytes); (2) evaluation of the levels of expression of eight monocyte functional markers, and (3) characterization of monocyte clusters defined through the dimensionality reduction of flow cytometry data (56 different clusters). The monocyte phenotype of the onART group closely resembled that of the UC group. In contrast, LTECs exhibited important alterations in the monocyte phenotype compared to that of the UCs, including (a) an increased proportion of intermediate monocytes and a decreased proportion of classical monocytes (*p* < 0.01), (b) altered expressions of functional markers across monocyte subsets (*p* < 0.05), and (c) alterations in sixteen different monocyte clusters (twelve decreased and four increased, *p* < 0.05). Many of these alterations were also observed when comparing the LTEC and onART groups. Our findings suggest that monocyte-driven mechanisms may contribute to HIV control in LTECs; however, some of these alterations could also promote systemic inflammation and immune activation. These observations provide a compelling rationale for considering therapeutic interventions in this unique population of PLWHs.

## 1. Introduction

Human immunodeficiency virus (HIV) infection exemplifies a persistent viral infection with no known model of spontaneous eradication. However, a subset of people living with HIV (PLWHs) known as elite controllers (ECs) possess the remarkable ability to suppress viral replication to undetectable levels in the absence of antiretroviral therapy (ART), making them the closest model to this scenario [1]. ECs are considered as the best example of a functional cure, in which the virus remains significantly suppressed but not fully eradicated, allowing individuals to maintain immunological and virological stability without the need for lifelong antiretroviral therapy (ART) because of their exceptional capacity to control HIV replication [1,2].

Despite this unique ability, a significant proportion of the EC population eventually loses viral control over time, either through a resurgence of viral replication or by immunological decline characterized by CD4 T-cell loss [3]. Only a small fraction of ECs maintains long-term infection control, the so-called long-term elite controllers (LTECs) [4]. Notably, several studies have shown that despite effective HIV control, ECs experience higher incidences of different comorbidities compared to those in the general uninfected population and to HIV patients with ART-mediated viral suppression [5,6]. This increased morbidity has been associated with persistent immune activation and systemic inflammation in ECs. Higher levels of T-cell activation have been observed in ECs compared to HIV-uninfected individuals and PLWHs on successful ART [7,8]. Additionally, ECs exhibit elevated levels of soluble inflammatory markers, including D-dimer and soluble tissue factors, soluble CD163, and various proinflammatory cytokines [5,9,10].

The available evidence suggests that the innate immune system plays a central role in driving immune activation and systemic inflammation in ECs [11]. Among the key cellular components of innate immunity, the monocyte/macrophage lineage has received particular attention [12,13]. Monocytes, a subset of myeloid-lineage cells, are characterized by their high phagocytic and proinflammatory capacities, as well as their ability to regulate adaptive T-cell responses [14]. Moreover, they serve as macrophage and dendritic cell precursors, two crucial cell types in HIV infection pathogenesis [12,15]. During acute infection, these cells exhibit phenotypic and functional alterations [13], including diminished phagocytic capacity [16], reduced type I IFN and γIFN production [17], and the expansion of CD14+CD16− classical monocytes lacking the expression of CCR2, which can suppress the CD8 T-cell response [18]. Key roles of monocytes in HIV disease pathogenesis are their contributions to persistent inflammation and immune activation [19]. Among the monocyte-associated disturbances linked to this phenomenon is the expansion of non-classical (CD14+CD16+) monocytes [20], which exhibit an increased capacity to produce inflammatory cytokines [21] and are correlated with high levels of soluble inflammatory markers [22].

Despite extensive research on the roles of monocytes in the pathogenesis of HIV infection, there are few studies specifically focusing on monocytes and ECs. Given that LTECs represent the closest model of HIV functional cure, research aimed at identifying the immunological correlates of protection against HIV disease should focus on this rare population of PLWHs. The objective of the present study was to perform a comprehensive characterization of monocyte phenotypes in a well-defined LTEC cohort and compare them to both non-controller PLWHs with uncontrolled viral replication and PLWHs with ART-mediated viral suppression.

## 2. Results

### 2.1. Characteristics of the Study Groups

A total of 67 volunteers were included in this study: 20 HIV-seronegative volunteers (the UC group) and 47 PLWHs. Among the PLWH participants, 10 were offART, 15 were onART, and 22 were LTECs. The characteristics of each group at inclusion are summarized in Table 1. All the groups were comparable in age, but there were differences in sex distribution: although most participants in the onART and offART groups were male, the LTEC and UC groups had a more balanced male-to-female ratio. CD4 counts were similar across the PLWH groups. The offART group presented the shortest time from diagnosis. Among the LTEC participants, the EC status had been maintained for a median of thirteen years, whereas in the onART group, ART had been administered for a median of six years.

### 2.2. Levels of Monocytes and Monocyte Subsets

Manual gating of flow cytometry data (using FCS Express v.7) was performed to analyze the total monocyte levels (as defined in the Materials and Methods Section 4) for each study participant. Additionally, according to CD14 (bright or dim) and CD16 (bright, dim, or negative) expression levels, four distinct monocyte subsets were defined and analyzed in each individual. Figure 1 presents the levels of the total monocytes and the four monocyte subsets across the study groups. The level of the total monocytes was similar across all the study groups. However, three of the four monocyte subsets presented significant differences across the groups (Kruskal–Wallis test, *p* < 0.05). Notably, the LTEC group displayed the most pronounced alterations, with significant deviations from the UC group in three of the four subsets. Specifically, compared to the UC reference group, the LTEC group showed significantly decreased levels of classical monocytes and CD14dCD16n monocytes (*p* = 0.012 and *p* = 0.002, respectively), and significantly increased levels of intermediate monocytes (*p* = 0.001) (Figure 1). These alterations were also significant when compared to those in the offART and onART groups, except for the CD14dCD16n subset, which showed similar levels in the LTEC and offART groups. In contrast, the onART group showed no significant differences in monocyte subset levels compared to those in the UC reference group, whereas the offART group exhibited a significant decrease only in the level of the CD14dCD16n subset (*p* = 0.019) (Figure 1).

### 2.3. Profound Alterations in Single Marker Expression Profiles in the LTEC Group

Next, using manual analysis with FCS Express v.7, the expression levels of eight different markers (CD11b, CD163, HLADR, SLAN, CCR2, CCR5, CXCR4, and CX3CR1) were compared between the PLWH groups and the UC reference group. Appendix A illustrates the single-marker expression profile for different monocyte subsets in the UC group. Interestingly, the LTEC group exhibited the most pronounced alteration of single-marker expression across the monocyte subsets (Figure 2 and Appendix A). The description of significantly altered markers in the LTEC group is as follows. Markers: Classical monocytes: Decreased CD11b and SLAN expressions;Intermediate monocytes: Decreased SLAN and increased CCR2 expressions;Non-classical monocytes: Decreased SLAN and increased CD11b, CCR2, and CCR5 expressions;CD14⁺CD16⁻ monocytes: Increased HLA-DR and decreased CD163 and CX3CR1 expressions.

Moreover, the LTEC group also presented significant changes in the mean fluorescence intensities (MFIs) for several markers, including decreased CD11b expressions in classical and intermediate monocytes; increased CD163 and/or HLADR expression(s) in classical, intermediate, and non-classical monocytes; and increased CCR5 and/or CCR2 expression(s) in intermediate and non-classical monocytes (Appendix A). In contrast, the offART group displayed fewer alterations in single-marker expressions, including decreased CD11b expressions in intermediate and non-classical monocytes and increased HLA-DR expression in CD14^+^CD16^−^ monocytes. Additionally, MFI analysis revealed increased HLA-DR and decreased CXCR4 expressions in classical, intermediate, and non-classical monocytes. No significant alterations were detected in the onART group.

### 2.4. Unsupervised Multidimensional Analysis of Flow Cytometry Data Identifies Monocyte Clusters That Distinguish Groups

Spectral flow cytometry analysis was performed to assess ten different monocyte markers: CD14 and CD16 (discriminating between negative, dim, and bright expressions); CD163 (scavenger receptor marker); HLADR (activation markers), CD11b (tissue-homing marker); SLAN (proinflammatory non-classical monocyte markers); and CCR2, CCR5, CXCR4, and CX3CR1 (chemokine receptors). To reduce the dimensionality and visualize the data distribution, t-distributed stochastic neighbor embedding (tSNE) was applied, generating a two-dimensional representation. The normalized density of the events across this tSNE map differed among the four study groups, as did the distribution of the events from each group (Appendix A). These findings suggest distinct monocyte phenotypes across the study groups.

Clustering analysis using the Louvain method identified 56 distinct clusters (from C00 to C55; Appendix A) differentiated by marker expression levels (Appendix A). Of these clusters, twenty-nine clusters belonged to the classical monocyte subset (CD14bCD16n), five clusters belonged to the non-classical monocyte subset (CD14dCD16b), twelve clusters belonged to the CD14dCD16n subset, and ten clusters contained a mixture of intermediate (CD14bCD16d) and classical or non-classical monocytes (Appendix A). The relative abundance of each cluster within the total monocyte population was calculated for each study sample, and differential abundance analysis was performed using the UC group as the reference. Several clusters were differentially expressed between the PLWH groups and the UC group (Figure 3 and Appendix A). Heatmaps depicting fold-change expressions in individual samples relative to UC medians revealed a distinct separation between the PLWH and UC groups, with the most pronounced differences observed in the LTEC group (Appendix A).

### 2.5. LTECs Show the Most Profound Alterations in Monocyte Clusters

Surprisingly, the LTEC group presented the highest number of differentially expressed clusters compared to the UC group, with sixteen clusters affected—twelve decreased, and four increased. In contrast, only four clusters were differentially expressed in the offART group and one in the onART group (Figure 3). Among the twelve decreased clusters in the LTEC group, nine belonged to the classical monocyte subset (CD14bCD16n), while the remaining three belonged to the CD14dCD16n subset. Of the four increased clusters, three were composed of classical and intermediate monocytes (clusters C20, C24, and C29), and the remaining one was formed from a mix of intermediate and non-classical monocytes (cluster C36).

Comparisons between the LTEC and other PLWH groups further defined a distinct monocyte cluster profile as follows:-LTECs vs. offART: A total of nine clusters were differentially expressed (seven decreased and two increased). All but one of the decreased clusters belonged to the classical monocyte subset, while the increased clusters belonged to the intermediate monocyte subset. Four of these clusters (C10, C20, C16, and C26) were also differentially expressed in LTEC vs. UC comparisons;-LTECs vs. onART: A total of thirteen clusters were differentially expressed (eleven decreased and two increased). Seven decreased clusters were from the classical monocyte subset, and four were from the CD14dCD16n subset, while the two increased clusters belonged to the intermediate monocyte subset. All but one of these clusters were also significantly altered in LTEC vs. UC comparisons (Figure 4).

## 3. Discussion

To the best of our knowledge, this is the first study to perform a detailed phenotypic analysis of monocytes in long-term elite controller PLWHs (LTECs), compared with both non-controller PLWHs with uncontrolled viral replication and non-controllers with treatment-suppressed viral replication, aiming to identify monocyte phenotypic markers associated with the LTEC status. Our key findings include the following: (a) The comprehensive phenotypic and multidimensional analyses revealed distinct differences in monocyte profiles across the PLWH groups. (b) The greatest alteration in the monocyte phenotype was observed in PLWHs with spontaneous control of viral replication. (c) The LTEC-specific monocyte profile was characterized by (1) an expansion of the intermediate monocyte subset accompanied by contractions of both classical and CD14dCD16n monocytes; (2) multiple phenotypic alterations, especially within the non-classical subset; and (3) significant changes in various monocyte clusters spanning the classical, intermediate, and CD14dCD16n subsets.

Although PLWHs with uncontrolled viral replication (the offART group) did not exhibit significant alterations in the overall distribution of the monocyte subpopulations (classical, intermediate, non-classical, and CD14dCD16n), we observed significant reductions in two clusters of classical monocytes (C05 and C34) and significant increases in one cluster of intermediate monocytes (C12) and one of CD14dCD16n monocytes (C25). These findings are consistent with previous reports indicating that alterations in the monocyte subset distribution occur early in acute HIV infection [23] and persist during chronic untreated infection [21,23,24,25]. Moreover, we found significant increases in the level of expression of the activation marker HLADR across all the monocyte subsets in offART PLWHs, which concurs with the current literature [25,26,27,28]. An interesting observation was the significant reduction in CXCR4 expression in all the monocyte subsets—except for the CD14dCD16n subset—in the offART group, given that lower CXCR4 monocyte expression has been associated with subclinical atherosclerosis [29].

In PLWHs with ART-mediated viral suppression, the monocyte phenotypic profile exhibited minimal alterations, with only an increased level of cluster C25 (a CD14dCD16n monocyte cluster with low expressions of CD11b and HLADR). This is in line with earlier reports indicating a trend toward the normalization of monocyte phenotypes in PLWHs on ART [26,30], although some studies have reported persistent changes in the monocyte subset distribution in PLWHs regardless of ART use [25].

The most intriguing results of our study are those found in the LTEC group. Studies addressing monocyte phenotypes in the EC population are very scarce, and to the best of our knowledge, this is the first investigation focusing on the population with sustained viral control. Despite complete viral suppression comparable to that of onART PLWHs, LTECs exhibited a broad array of phenotypic monocyte alterations. The various analytical approaches applied to the cytometry data consistently revealed distinct modifications in monocyte phenotypes. First, among the four monocyte subsets defined by CD14 and CD16 expressions, LTECs displayed a significant increase in the intermediate subset alongside decreases in both the classical and CD4dCD16n monocyte subsets. Second, LTECs exhibited increases in several clusters of intermediate monocytes and decreases in several clusters of classical monocytes, consistent with the overall subset findings. Finally, alterations were detected in the expressions—both in positivity and intensity—of several single markers across different monocyte subsets. Given the dual role of monocytes in HIV pathogenesis [31,32,33], these findings may indicate that the observed phenotypic alterations contribute either beneficially, by promoting spontaneous viral control, or detrimentally, by fostering persistent immune activation and systemic inflammation in ECs [5,9,10].

A key finding in the LTEC population was the marked expansion of the intermediate monocyte subset, which was increased not only compared to that of the uninfected control group but also compared to those of the offART and onART groups. As intermediate monocytes are primarily responsible for antigen processing and presentation to adaptive immune cells [34], the expansion of this population in LTECs raises the possibility of this subset contributing significantly to the enhanced HIV-specific T-cell response observed in ECs [35]. In line with this hypothesis, previous studies have reported that myeloid dendritic cells in ECs exhibit a markedly increased antigen-presenting capacity compared to those in non-controllers [36]. Conversely, the increase in intermediate monocytes may also be linked to heightened inflammation and elevated risk of developing various comorbidities, as several studies have associated this subset with adverse clinical outcomes in PLWHs [37,38,39]. Another noteworthy observation in LTECs was the decreased level of the CD14dCD16n monocyte subset compared with those in both the UC and the onART groups. This subset has been scarcely investigated in the literature; however, one study reported that lower levels of this monocyte subpopulation are associated with subclinical cardiovascular disease (sCVD) in PLWHs undergoing treatment, suggesting a potential protective effect [40].

In addition to changes in the subset distribution, the expressions of several markers in different monocyte subsets were markedly altered in the LTEC population. CD11b expression—assessed as the percentage of positive cells and/or as the intensity of the expression—was reduced in classical and intermediate monocytes. Because CD11b plays a critical role in monocyte migration under both physiological and pathological conditions [41] and is involved in the HIV infection of monocytes/macrophages [42] as well as in proinflammatory cytokine production, the decreased expression of CD11b in LTECs may lead to reduced sensitivity to CD11b-mediated activation, resulting in a lower proinflammatory response [43,44]. Similarly, SLAN expression was reduced in all the monocyte subsets except for the CD14dCD16n subset. Given that SLAN is predominantly expressed in non-classical monocytes—[45] which are elevated in viremic PLWHs and possess a high capacity for TNFa production—the diminished SLAN expression in LTECs could beneficially limit immune activation and viral replication.

In contrast to CD11b and SLAN, the chemokine receptors CCR2 and CCR5 were markedly upregulated in intermediate and non-classical monocytes in LTECs. This finding is particularly unexpected for non-classical monocytes, a subset that typically exhibits low or negligible expression of chemokine receptors. Median levels of CCR2 and CCR5 expression in non-classical monocytes were several times higher in LTECs compared with the other groups. Considering the role of these receptors in the monocyte inflammatory response [46,47], these data suggest that these monocyte populations may contribute to the chronic inflammation observed in ECs [5,6]. Elevated CCR2 expression in monocytes has been linked to conditions such as renal dysfunction [48], cardiovascular disease [49], or HIV-associated neurocognitive disorders (HANDs) [50].

Furthermore, increased HLADR expression was observed in all the LTEC monocyte subsets, except for the intermediate subset. This pattern, which resembles that seen in patients with high levels of viral replication and in contrast to findings in patients with ART-mediated viral suppression, may indicate persistent low-level viral replication in LTECs sufficient to induce heightened HLADR expression. Alternatively, given the role of HLADR in antigen presentation [51], these findings might suggest an enhanced capacity for antigen presentation in LTEC monocytes, potentially contributing to viral control [52]. Finally, CD163 expression was elevated in all the LTEC monocyte subsets except for CD14dCD16n. As sCD163 is regarded as a marker of monocyte activation [53], and its soluble form in plasma correlates with adverse clinical outcomes in PLWHs [54,55,56], the increased CD163 expression in LTECs could reflect ongoing monocyte stimulation because of residual HIV replication or microbial translocation [5,57]. Alternatively, increased expression of CD163 might represent a compensatory anti-inflammatory response [58,59].

Finally, the high inter-individual variability in the classical and intermediate monocyte subsets observed in the LTEC group, as opposed to the more uniform distribution of values in these monocyte subsets in the other PLWH groups, is a finding that warrants discussion. As an explanation for this greater variability, we hypothesize that LTEC patients are a rather heterogeneous group [3] in terms of factors such as the time of the infection control, CD4 count, and other unknown factors that may be related to the various mechanisms operating in these patients to achieve the control of viral replication and infection progression. This could explain the greater variability in monocyte populations observed in LTEC patients. 

It is important to acknowledge several limitations of our study. First, the cross-sectional design precludes the evaluation of the temporal evolution and stability of the monocyte phenotype. Second, the relatively small sample size limits the robustness of our findings. Third, the comparisons between the offART and onART groups with the rest of the groups may be biased by the different distribution of sexes (predominantly male in the offART and onART groups and a more balanced distribution of male and female in the LTEC and UC groups); however, we examined how sex affected the levels of monocyte subsets and clusters and found no discernible sex-based differences. Fourth, we did not evaluate the monocyte function, which could support some of the theories discussed, like enhanced antigen presentation or the generation of various cytokines in response to stimulation. However, we were unable to conduct such analyses because of the limited sample availability. Finally, we were unable to assess the relationship between the monocyte phenotype and the degree of systemic inflammation or morbidity burden within the study population, as these parameters were not examined.

In conclusion, our study is the first to report extensive alterations in monocyte phenotypes in PLWHs with long-term control of HIV (LTECs) compared with those exhibiting ART-mediated viral suppression. The diverse array of monocyte phenotypic abnormalities observed in the LTEC population suggests the existence of monocyte-driven mechanisms underlying HIV control; however, some alterations may also contribute to systemic inflammation and immune activation. Further studies are warranted to confirm the potential association between these monocyte phenotypic changes and systemic inflammation and morbidity in this unique PLWH population.

## 4. Materials and Methods

### 4.1. Study Design and Participants

Adult participants with chronic HIV infection (people living with HIV or PLWH groups) and HIV-seronegative volunteers (uninfected controls or the UC group) were included in this cross-sectional study. The PLWH groups comprised long-term elite controllers (the LTEC group), defined as PLWHs who had maintained spontaneous control over their HIV infection for at least five years throughout their follow-up period; non-controller PLWHs on antiretroviral therapy (onART), with an undetectable plasma HIV viral load (pVL) at the time of the study inclusion; and non-controller PLWHs off ART (the offART group), with a detectable HIV pVL. The participants in the LTEC group were selected from the Spanish ECRIS database, a previously reported multicenter cohort of PLWH elite controllers (ECs) initiated in 2013 [60]. This cohort includes participants from the AIDS research cohort (CoRIS), the long-term non-progressor (LTNP) cohort, and several clinical centers in Spain (see the Appendix A). All the participants provided written informed consent. This study was approved by the ethical review board of the Instituto de Investigación Sanitaria-Fundación Jiménez Díaz, Madrid, Spain (approval ID: PIC097-19_FJD) and was conducted in accordance with the Declaration of Helsinki.

### 4.2. Cell Samples

Blood samples were collected by venipuncture in tubes containing ethylenediaminetetraacetic acid (EDTA) as an anticoagulant and were sent on the same day to the Spanish HIV HGM BioBank (http://hivhgmbiobank.com/?lang=en (accessed on 7 April 2025)). The samples were immediately processed to obtain peripheral blood mononuclear cells (PBMCs) by gradient centrifugation using ficoll-hypaque solution (Sigma-Aldrich, Burlington, MA, USA) and were subsequently cryopreserved in liquid nitrogen. The PBMC samples were kindly provided by the Spanish HIV HGM Biobank.

### 4.3. Multiparameter Flow Cytometry

One million PBMCs were stained with a panel of thirteen different monoclonal antibodies (plus a viability dye) (see Appendix A). A detailed staining protocol is provided in the Appendix A. Following staining, cells were acquired using an Aurora spectral flow cytometer (Cytek Biosciences, Freemont, CA, USA) with a minimum of 200,000 events collected per sample for subsequent analysis. The data were analyzed using FCS Express v.7 (DeNovo software, Pasadena, CA, USA). Dead cells were excluded using the live/dead viability dye, and live monocytes were gated based on forward (FSC) and side (SSC) scatter properties; single monocytes were then selected using the FSC width versus the FSC height. From the single live monocyte population, sequential gating was performed to exclude B-cells (CD19+) and NK cells (CD56+), and CD14+ cells (monocytes) were selected for further analysis. According to the relative expression levels of CD14 (bright (b) and dim (d)) and CD16 (negative (n), dim (d), and bright (b)), four monocyte subsets were defined: CD14bCD16n (classical monocytes), CD14bCD16d (intermediate monocytes), CD14d/bCD16b (non-classical monocytes), and CD14dCD16n monocytes. An example of gating is provided in Appendix A.

### 4.4. Unsupervised Analysis of Flow Cytometry Data

A detailed protocol for the unsupervised analysis of the flow cytometry data is provided in the Appendix A (see Appendix A).

### 4.5. Statistical Analysis

The clinical and epidemiological characteristics of the different study groups were expressed as medians with interquartile ranges. The differences between the groups were evaluated using non-parametric tests (e.g., the Kruskal–Wallis test or Mann–Whitney test, as appropriate). The levels of the different monocyte clusters were compared across the study groups, as detailed in the Appendix A.

## Figures and Tables

**Figure 1 ijms-26-03926-f001:**
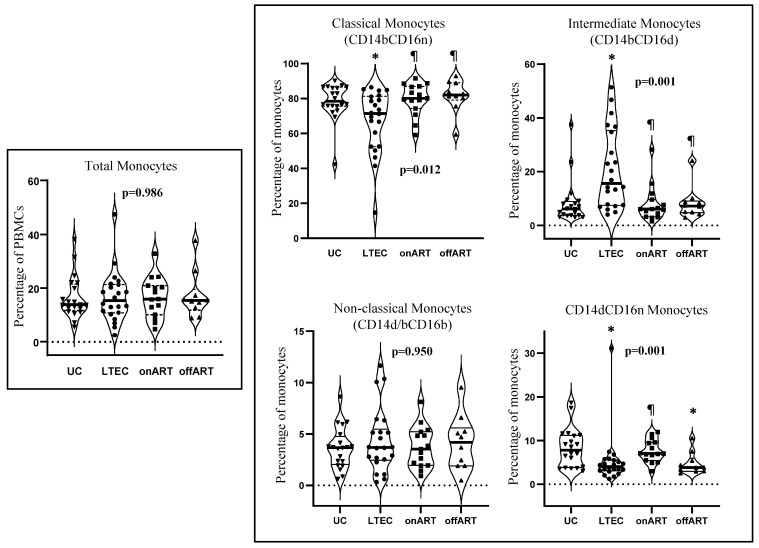
Violin-plot graphs showing the levels of monocytes (left graph) and of different subsets of monocytes based on the relative expressions of CD14 and CD16 markers (right graphs) in the different study groups. The Y-axis on the graphs represents the level expressed either as the percentage of peripheral blood mononuclear cells (for the left graph showing the total monocytes) or as the percentage of the total monocytes (for the right graphs showing the different subsets of monocytes). The *p*-values inside the graphs are for the comparison among the four study groups (Kruskal–Wallis test); (*) *p* < 0.05 compared to uninfected controls (the UC group); (¶) *p* < 0.05 compared to the LTEC group (Mann–Whitney U test). CD14 and CD16 expressions were categorized into three levels: negative (n), dim (d), and bright (b).

**Figure 2 ijms-26-03926-f002:**
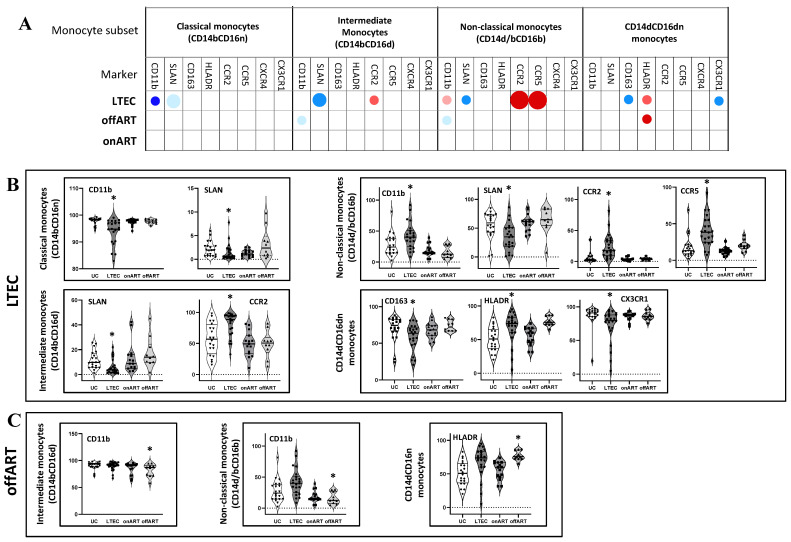
(**A**) Schematic representation (bubble diagram) of the profile of the alterations in single-marker expressions by different subsets of monocytes in the PLWH groups compared to the UC group as a reference. Each dot in the table represents a significant difference with respect to the UC reference group. The size of the dot indicates the degree of the difference in the fold change: <2, 2–3, and >3 from the smallest to the biggest dots. Red colors indicate increases, and blue colors indicate decreases with respect to the UC group. The level of statistical significance (corrected *p*-value) is indicated by the color tone: 0.05–0.01, 0.01–0.001, and <0.001 for the light, medium, and dark tones, respectively. (**B**) Violin plots of the expression levels of each single marker by different monocyte subsets in the LTEC study groups: (*) *p* < 0.05 with respect to the UC group. Only those single markers and monocyte subsets that showed a significant difference between the LTEC and UC groups are shown. (**C**) Violin plots of the expression levels of each single marker by different monocyte subsets in the offART study groups: (*) *p* < 0.05 with respect to the UC group. Only those single markers and monocyte subsets that showed a significant difference between the offART and UC groups are displayed.

**Figure 3 ijms-26-03926-f003:**
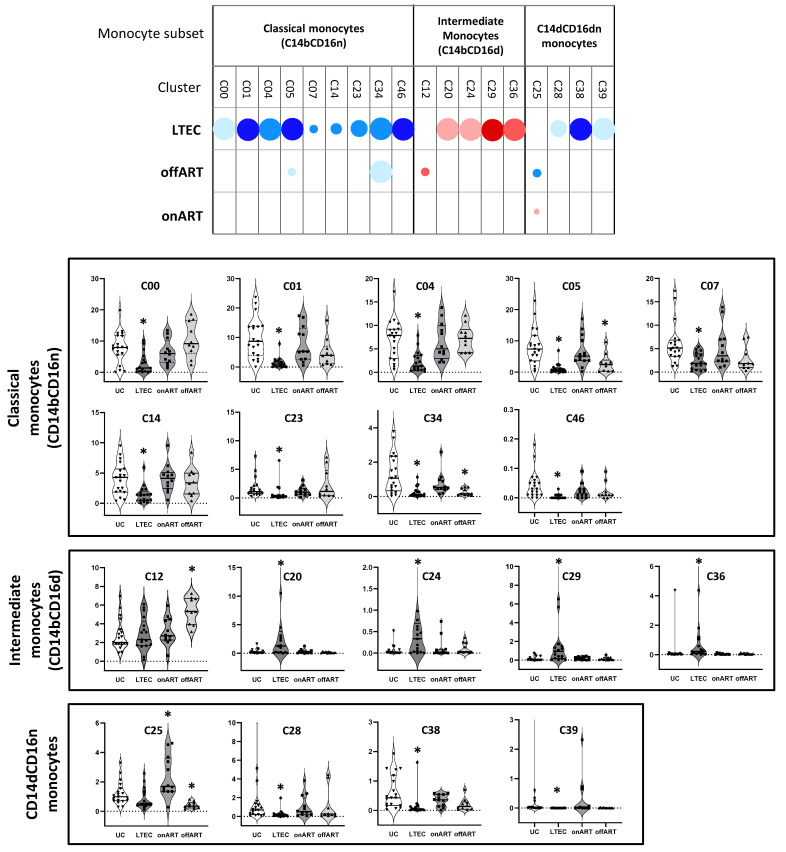
The upper part of the figure shows a schematic representation (bubble diagram) of monocyte clusters with significant differences between the PLWH groups and the UC group. Clusters are grouped according to CD14 and CD16 expressions into different monocyte subsets. Each dot in the table represents a significant difference (adjusted *p* < 0.05) with respect to the UC group. The size of the dot (from the smallest to the biggest size) indicates the degree of the difference in the fold change: <2, 2–3, 3–4, 4–5, and >5. Red colors indicate increases, and blue colors indicate decreases with respect to the UC group. The level of statistical significance (corrected *p*-value) is indicated by the color tone: 0.05–0.01, 0.01–0.001, and <0.001 for the light, medium, and dark tones, respectively. The lower parts of the figure show violin-plot graphs of the levels (expressed as percentages over the total monocytes) of each cluster in the four study groups. As in the upper part, clusters are grouped according to the subset of the monocytes: (*) adjusted *p* < 0.05 compared to the UC group.

**Figure 4 ijms-26-03926-f004:**
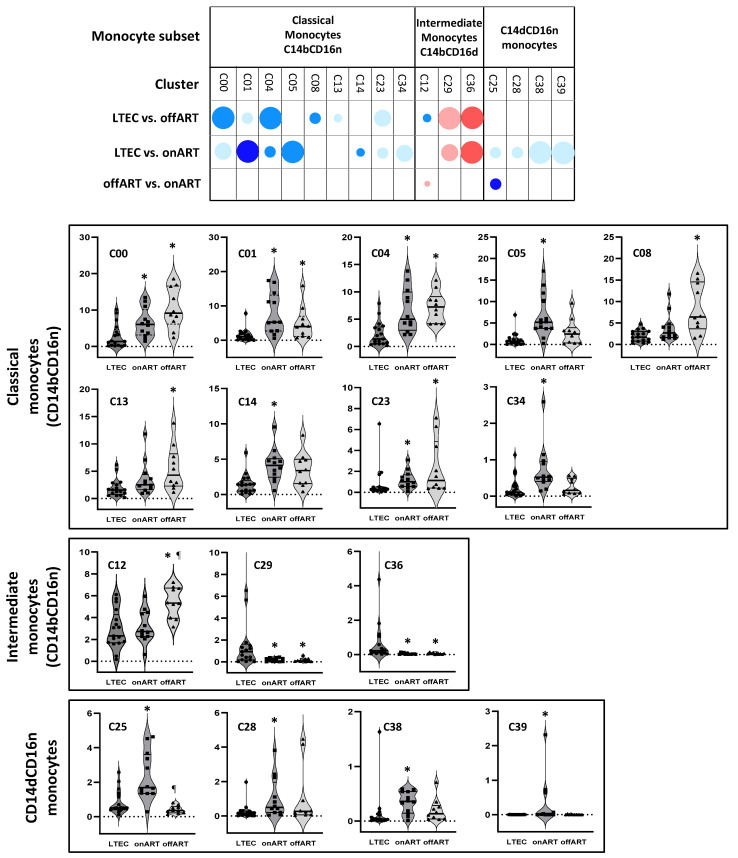
The upper part of the figure shows a schematic representation (bubble diagram) of monocyte clusters with significant differences between pairs of PLWH groups. Clusters are grouped according to CD14 and CD16 expressions into different monocyte subsets. Each dot in the table represents a significant difference with respect to the reference group, as indicated in the figure. The size of the dot indicates the degree of the difference in the fold change: <2, 2–3, 3–4, 4–5, and >5 from the smallest to the biggest dot. Red colors indicate increases, and blue colors indicate decreases with respect to the reference group. The level of statistical significance (corrected *p*-value) is indicated by the color tone: 0.05–0.01, 0.01–0.001, and <0.001 for the light, medium, and dark tones, respectively. The lower parts of the figure show violin plots of the levels (expressed as percentages over the total monocytes) of each cluster in the three PLWH groups. As in the upper part, clusters are grouped according to the monocyte subset: (*) *p* < 0.05 with respect to the LTEC group; (¶) *p* < 0.05 with respect to the onART group.

**Table 1 ijms-26-03926-t001:** Characteristics of groups included in this study.

Characteristic	LTEC (n = 22)	onART (n = 15)	offART (n = 10)	UC (n = 20)	*p*-Value
Age (years)	44 [35–49]	44 [42–49]	43 [36–49]	43 [36–49]	0.266
Sex (% of males)	59	80	100	50	**0.025**
Years since HIV diagnosis	15 [6–20]	12 [5–15]	5 [3–9]	NA	**0.027**
Years as EC	13 [7–16]	NA	NA	NA	NA
Years on ART	NA	6 [3–9]	NA	NA	NA
Plasma HIV load (copies/mL)	50	50	57937 [28,817–80,676]	NA	NA
CD4 count (cells/μL)	837 [603–1210]	820 [599–1127]	625 [518–920]	NA	0.200

Data are given as median [Q1–Q3], except sex that is expressed as percentage. Statistical differences between the four study groups were tested by Kruskall-Wallis test for continuous variables and by Chi-square test for sex. Statistical significant was considered for *p*-values below 0.05 (in bold in the table). NA: not apply.

## Data Availability

The raw data supporting the conclusions of this article are available on request from the corresponding author.

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
