# Peer review of "Long-Term Elite Controllers of HIV-1 Infection Exhibit a Deep Perturbation of Monocyte Homeostasis"

_ijms, 2025, doi:10.3390/ijms26093926_

Round 1

Reviewer 1 Report

Comments and Suggestions for Authors

The aim of the research is highly relevant. However, the manuscript has several limitations and flaws that preclude its publication.

  1. The "flow cytometry example" in Supplementary Figure 7 shows an incorrect gating strategy, raising doubts about all the results. The values on the X and Y axes are omitted; not necessarily incorrect, but it makes interpretation difficult. In the first dot plot, the "live cells" gate includes only live cells, which is highly unusual for cryopreserved samples and suggests incorrect gating. The subsequent monocyte gate is strange, are you sure the axis labels are correct? Did you apply a threshold to SSC? While not incorrect, this is unusual. The singlet gating is also strange. Additionally, you show an LTEC population with approximately 50% monocytes, are you certain about this value?
    1. How did you define CD16+ cells? Did you use FMO controls or other gating controls? This could significantly impact your results. The gating strategy used to define the four monocyte populations appears incorrect. To accurately gate CD16+ cells, proper control is necessary.
    2. How did you determine positive gates for CD11b, SLAN, CD163, HLA-DR, CCR2, CCR5, CXCR4, and CXCR1? In the CD56- population, it seems you are also including CD56+ cells.
  • Overall, all flow cytometry analyses need revision, which means all results will be affected.
  1. How do you define Long-Term Elite Controllers (LTEC)? Your LTEC group appears highly heterogeneous, with some participants not meeting the expected duration criteria. This heterogeneity could explain the variation in monocyte subsets within the group. A similar pattern is observed in CD4 counts.
  2. Why are the "years since HIV diagnosis" and "years as EC" different?
  3. The proportion of sexes differs—did you adjust your analysis using sex as a confounding factor?
  4. Your statistical analysis needs improvement. How did you handle outliers?
  5. The figure resolution is low.
  6. You need to add Y-axis labels in Figure 1.
  7. In the Supplementary Material, the phrase "excluding CD3, CD16, and CD56, which were used to exclude contaminating T cells, B cells, and NK cells, respectively" should be corrected, CD16 should be replaced with CD19.
Comments on the Quality of English Language

The English could  be improved for better readability.

Author Response

Comment 1: The "flow cytometry example" in Supplementary Figure 7 shows an incorrect gating strategy, raising doubts about all the results. The values on the X and Y axes are omitted; not necessarily incorrect, but it makes interpretation difficult. In the first dot plot, the "live cells" gate includes only live cells, which is highly unusual for cryopreserved samples and suggests incorrect gating. The subsequent monocyte gate is strange, are you sure the axis labels are correct? Did you apply a threshold to SSC? While not incorrect, this is unusual. The singlet gating is also strange.

Response: we agree with the reviewer that supplementary Figure 7 has some shortcomings and accordingly we have modified this figure to make it clearer: a) we have improved resolution; b) we have added the values on the X and Y axes; c) we have corrected some gates (gate to exclude CD56+ cells). Regarding the comments of the reviewer on the correctness of gating strategy, we sincerely think that the strategy is adequate. The first dot-plot (FSC-A versus Live/dead) shows all the events acquired (ungated; we have added this information in the revised figure for clarity). There are almost no positive events for Live/dead (Y axis on the plot), because in our hands the viability of thawed PBMCs is always greater than 95%. This is because every step of the freezing and thawing process for PBMCs is meticulously planned and carried out according to the most stringent guidelines. The singlet gating is also OK and shows that the majority of events are singles, as it is expected if the cell sample has been well re-suspended. For monocyte gating, several sequential gating steps were performed (as explained in the figure legend). First, contaminating CD3+ (T cells) events were excluded; second CD19+ (B cells) events were excluded; third CD56+ events (NK cells) were excluded and lastly only CD14+ (monocytes) were selected for further analysis. We did not apply any SSC threshold. Using this sequential strategy we ended with a highly pure population of monocytes, defined as a population of PBMCs based not only on FSC and SSC parameters but also on the exclusion of contaminating CD3+, CD19+ and CD56+ cells.

Comment 2: Additionally, you show an LTEC population with approximately 50% monocytes, are you certain about this value?. 

Response: As shown in supplementary figure 7, the proportion of monocytes over the total population of PBMCs is 24.6%, based on FSC and SSC criteria. After applying the sequential gating strategy to exclude CD3+, CD19+ and CD56+ events, this proportion is slightly lower (24.6x0.978x0.993x0.986x0.997 = 23.5%)

Comment 3: How did you define CD16+ cells? Did you use FMO controls or other gating controls? This could significantly impact your results. The gating strategy used to define the four monocyte populations appears incorrect. To accurately gate CD16+ cells, proper control is necessary.

Response: Yes, we used FMO controls for every marker used in the staining panel. During the setting up process, and FMO (fluorescence minus one) was performed to set up the threshold to separate positive from negative events for every marker. Regarding the gating to differentiate between different subsets of monocytes (classical, intermediate, non-classical and CD14dimCD16neg monocytes), we followed the phenotypic definition (based on CD14 and CD16 markers) of classical, intermediate and non-classical monocytes already published in several previous studies (see review by Ziegler-Heitbrock et al; Blood 2010; 116(16):e74-78; doi: 10.1182/blood-2010-02-258558). A fourth subset of monocytes was also included in the analysis (CD14dimCD16neg monocytes) given that this subset also met criteria of monocytes as previously explained in the gating strategy (see Barbour et al Atherosclerosis 2014; doi: 10.1016/j.atherosclerosis.2013.10.021; Villani et al Science 2017; 356(6335) doi:10.1126/science.aah4573; and also McCausland et al PLOS ONE 2015; 10(10):e0139474 doi:10.1371/journal.pone.0139474)

Comment 4: How did you determine positive gates for CD11b, SLAN, CD163, HLA-DR, CCR2, CCR5, CXCR4, and CXCR1? In the CD56- population, it seems you are also including CD56+ cells.

Response: As we have commented before (response to comment #3) we performed an FMO to establish the threshold of positivity for every marker included in the staining panel. Regarding the gating of the CD56neg population (as shown in supplementary fgure 7) we agree with the reviewer that we were including also some CD56+ cells. This was a mistake and in the revised version of this figure this has been corrected (see also response to comment #1).

Comment 5: Overall, all flow cytometry analyses need revision, which means all results will be affected.

Response: We understand the reviewer concerns regarding the flow cytometry analysis. However, as we have explained in the response to comment #1, we think that the gating strategy and placement of gates for positive events for each marker are correct. Nonetheless we have revised the gating strategy and the analysis protocol used to extract the data for each sample and results are the same as those reported in the manuscript.

Comment 6: How do you define Long-Term Elite Controllers (LTEC)? Your LTEC group appears highly heterogeneous, with some participants not meeting the expected duration criteria. This heterogeneity could explain the variation in monocyte subsets within the group. A similar pattern is observed in CD4 counts.

Response: as explained in the Methods section of the manuscript (“Study design and participants” subheading), LTEC were defined as PLWH who had maintained spontaneous control over their HIV infection for a minimum of five years and throughout their whole follow-up period. We agree with the reviewer that the population of LTEC participants is heterogeneous in terms of length of disease control and CD4 counts. However all of them met the inclusion criteria. The median length of disease control (as shown in Table 1) was 13 years, with a P25-P75 of 7-16 years, meaning that 25% of the population had a length of control between 5-7 years; 25% had a length of control above 16 years and 50% had a length of control between 7 and 16 years. Regarding CD4 counts, all LTEC had a count above 500 cells/microliter, with a median (as shown in Table 1) of 837 cells/microliter. As the reviewer points out, this heterogeneity could potentially explain the heterogeneity in monocyte subsets within the group of LTEC. However, we did not find any correlation between the levels of the various monocyte subsets and either the length of infection or the CD4 counts (data not shown in the manuscript), which suggests that the heterogeneity observed in monocyte subsets is independent of both of these factors (length of infection and CD4 counts). This new information has been added into the revised version of the manuscript (discussion section, paragraph 9).

Comment 7: Why are the "years since HIV diagnosis" and "years as EC" different?.

Response: This is due to the fact that we only considered "years as EC" for years in which measurements of plasma HIV viral load were available and below the detection limit. There were times (at the beginning of follow-up) when some participants' plasma VIH load was not measured; these times were not considered "periods of HIV control."

Comment 8: The proportion of sexes differs—did you adjust your analysis using sex as a confounding factor?.

Response: Given the variations in the distribution of sexes among the various study groups (as indicated in Table 1), we concur with the reviewer that sex may potentially introduce confounding variables into the analysis of monocyte subsets and clusters. However, we examined how sex affected these metrics and discovered no discernible sex-based differences. We have incorporated this information into the revised version of the manuscript (discussion section, paragraph 10).

Comment 9: Your statistical analysis needs improvement. How did you handle outliers?

Response: We used median and interquartile range as measures of central tendency and dispersion respectively for all continuous variables analyzed. Because it is the middle value in a data set, the median is less vulnerable to outlier distortion. A single outlier will have little to no impact on the median because it is dependent on the order of the values rather than their magnitude. Because of this, the median is a reliable indicator of central tendency in situations involving skewed distributions or data sets that contain outliers.

Comment 10: The figure resolution is low

Response: we agree with the reviewer that figures appearing in the old version of the manuscript are very difficult to read due to the very low resolution they show. Following reviewer suggestion, we have changed the format of figures appearing in the main text (figure 1, figure 2, figure 3, and figure 4) to increase their resolution and make them readable.

Comment 11: You need to add Y-axis labels in Figure 1.

Response: we have added Y-axis labels to Figure 1.

Comment 12: In the Supplementary Material, the phrase "excluding CD3, CD16, and CD56, which were used to exclude contaminating T cells, B cells, and NK cells, respectively" should be corrected, CD16 should be replaced with CD19.

Response: This was a mistake and in the revised version of the manuscript it has been corrected.

Comment 13: The English could  be improved for better readability.

Response: following the reviewer suggestion, a native English speaker has revised the manuscript to improve the readability.

Reviewer 2 Report

Comments and Suggestions for Authors

This study is the first to detail monocyte phenotypes in long-term elite controllers (LTEC) with HIV, comparing them to non-controllers with and without treatment. LTEC showed the greatest monocyte phenotype changes, including expanded intermediate monocytes and reduced classical and CD14dCD16n monocytes. Non-controllers with uncontrolled viral replication showed increased monocyte activation markers, while those with suppressed viral replication showed few alterations. LTEC’s altered monocyte phenotype may contribute to viral control or persistent inflammation. The expansion of intermediate monocytes, crucial for antigen presentation, could improve T-cell response, while also potentially increasing inflammation. Decreased CD14dCD16n monocytes might protect against cardiovascular disease. Changes in marker expression, such as reduced CD11b and SLAN, suggest reduced pro-inflammatory response. Non-classical monocytes in elite controllers showed significantly higher CCR2 and CCR5 expression than other groups. This suggests these monocytes contribute to chronic inflammation, as elevated CCR2 is linked to various pathologies.

Briefly, although there are limitations of this study as the authors claimed, the data was well presented and carefully analyzed. After careful consideration, I recommend accepting this paper for publishing in IJMS.

Author Response

Comment 1: Briefly, although there are limitations of this study as the authors claimed, the data was well presented and carefully analyzed. After careful consideration, I recommend accepting this paper for publishing in IJMS.

Response: we greatly appreciate the reviewer comment.

Reviewer 3 Report

Comments and Suggestions for Authors The current manuscript by Benito et al., titled “Long term elite controllers of HIV-1 infection exhibit a deep 2 perturbation of monocyte homeostasis” describes the phenotype of monocytes in a well-defined cohort of LTEC in comparison to non-controller PLWH with uncontrolled viral replication and PLWH with ART-mediated viral replication control.A total of 67 volunteers with twenty HIV-seronegative volunteers (UC group) and 47 PLWH volunteers were included in the study. Data shows that the total monocytes was similar across all study groups, however, of the four monocyte subsets, three of them presented global differences when comparing the four study groups. Here, the LTEC group showed significant decreased level of classical monocytes and CD14dCD16n monocytes, and significantly increased level of intermediate monocytes with alterations compared to off ART and on ART groups.A total of 13 clusters were differentially expressed, 11 decreased and 2 increased in LTEC. Finally, CD11b and SLAN markers, chemokine receptors CCR2 and CCR5 were highly increased in intermediate and non-classical monocytes of LTEC. Overall, I found the study very timely and novel, however, my concerns are: 1.There were no validation (i.e., PCRs) for any of the markers that were either up – or down regulated. 2.Type of virus (WT vs. mutants), copy numbers, or viral gene expression was completely missing in this study! 3.Functionality of monocyte subsets is missing, therefore not clear what is the significance of this current study!

Author Response

Comment 1: There were no validation (i.e., PCRs) for any of the markers that were either up – or down regulated.

Response: we agree with the reviewer that having an alternative method (such as PCR) to validate our findings regarding the expression level of phenotypic markers would strengthen our conclusions. Unfortunately, the cell sample availability precluded us from performing such assays.

However, we think that measuring the expression level of the protein, as we did using flow cytometry and the mean fluorescence intensity (MFI) parameter as a surrogate of protein level of expression, is a reliable enough technique to draw the conclusions discussed in the manuscript.

Comment 2: Type of virus (WT vs. mutants), copy numbers, or viral gene expression was completely missing in this study.

Response: all HIV patients included in the study were infected with HIV-1 subtype B. Regarding copy numbers or viral gene expression, the level of HIV-RNA copies (copy numbers of HIV virions) in plasma for each group of patients is reported in Table 1. For LTEC and patients on ART the level of HIV-RNA copies was undetectable using the commercial assays. However, the existence of very low levels of HIV-RNA copies in plasma of these two groups of patients (below the limit of detection of commercial assays) cannot be ruled out

Comment 3: .Functionality of monocyte subsets is missing, therefore not clear what is the significance of this current study.

Response: We concur with the reviewer that research examining monocyte functionality will enhance the phenotypic analysis in the manuscript and may help validate some of the theories put forth in the discussion, such as improved antigen presentation or the production of particular cytokines in response to stimulation. However, we were unable to conduct any of these functional studies because of the availability of cell samples. The updated version of the manuscript now includes this as a study limitation (discussion section, paragraph 10).

However, as discussed in the manuscript, we think that the findings of this study are significant and advance our knowledge of the pathophysiology of HIV infection by demonstrating the presence of significant phenotypic changes in monocytes from HIV patients with long-term spontaneous control of viral replication (LTEC). This suggests the critical role of these cells in aspects of pathogenesis, such as the mechanisms governing disease control or the mechanisms causing the persistent inflammatory state.

Reviewer 4 Report

Comments and Suggestions for Authors

This study revealed the perturbation of monocyte homeostasis by phenotypic analysis, the findings are very interested. I have following concerns.

Table 1 lists all 67 participants across the four groups, including their infection history, duration on EC/ART, age, and sex, which is great. However, some details are missing. Do these participants have other pathogenic conditions, such as comorbidities? If the answer is yes, how this will affect your results

In Figure 1, the LTEC group shows a wide range in classical, intermediate, and non-classical monocytes, with some individuals displaying levels similar to other groups and others showing notable differences. Additionally, one donor exhibits an extremely high level of CD14+CD16- monocytes. Please provide an explanation for this outlier.

The authors state that monocytes from the LTEC group are associated with an expansion of the intermediate monocyte subset. However, the data in Figure 1 show a wide range of levels across different donors within this group. It may be helpful to divide the LTEC group into subgroups and analyze potential contributing factors.

For Figure 2, the data presentation is unclear. Please add subfigure labels (e.g., Figure 2A, 2B, etc.) to each panel to help readers easily match the description with the figures. Additionally, in the lower part of the figure, you mention displaying data for the LTEC and OFFART groups, but each panel includes data from all four groups. Please clarify this inconsistency.

Overall, only performing a phenotypic study may not be sufficient to characterize monocyte perturbation in LTEC. I suggest conducting ex vivo studies to assess monocyte function, such as evaluating whether monocytes from LTEC exhibit increased antigen presentation or cytokine production etc..

Author Response

Comment 1: Table 1 lists all 67 participants across the four groups, including their infection history, duration on EC/ART, age, and sex, which is great. However, some details are missing. Do these participants have other pathogenic conditions, such as comorbidities? If the answer is yes, how this will affect your results.

Response: We did not examine the existence of comorbidities in the different study groups because this was not the main objective of our investigation. Indeed, as we note in the discussion, we hypothesize that the comorbidity profile may be influenced by the distinct profiles of phenotypic changes observed in LTEC patients as opposed to patients receiving antiretroviral therapy. Nevertheless, we have mentioned this as a limitation in the revised version of manuscript's discussion (discussion section, paragraph 10).

Comment 2: In Figure 1, the LTEC group shows a wide range in classical, intermediate, and non-classical monocytes, with some individuals displaying levels similar to other groups and others showing notable differences. Additionally, one donor exhibits an extremely high level of CD14+CD16- monocytes. Please provide an explanation for this outlier.

Response: We value this insightful observation of the reviewer. We concur that the classical and intermediate monocyte subpopulations exhibit the most interindividual variation in LTEC patients, while the non-classical monocyte population exhibits variation that is more comparable to that of the other study groups. With the exception of the single outlier in the value distribution, the interindividual variation in the CD14dCD16n monocyte subset in LTEC patients is also comparable to that of other groups. As an explanation for this greater variability, we hypothesize that LTEC patients are a rather heterogeneous group (see review by Navarrete-Muñoz et al. 2020; doi: 10.1080/21505594.2020.1788887; reference #3 of the manuscript) in terms of factors such as time of infection control, CD4 count, etc., and other unknown factors that may be related to the various mechanisms that may be operating in these patients to achieve control of viral replication and infection progression. This would explain the greater variability in monocyte populations observed in LTEC patients.

Regarding the outlier with a very high level of CD14dCD16n monocyte population, we have reviewed the flow cytometry analyses, and can confirm that the data for this outlier is correct. We have checked for the existence of potential differences between this donor and the rest of the LTEC group in terms of age, lenght of infection, and CD4 counts and found no significant differences. However it is interesting that, in this donor, the great expansion of CD14dCD16n subset (31% of total monocytes, compared with a median of 4.1% for the LTEC group) was also accompanied by an expansion of the intermediate monocyte subset (47% of total monocytes, compared with a median of 16% for the LTEC group) and a contraction of the classical monocyte subset (15% of total monocytes, compared with a median of 71% for the LTEC group). As we discuss in the manuscript, expansion of intermediate monocyte subset could be associated with better HIV-specific adaptive T cell responses, whereas the expansión of the CD14dCD16n subset could impact on cardiovascular morbidity (see Barbour et al Atherosclerosis 2014; reference #40). However, neither the level of virus-specific T cell reponses nor the presence of cardiovascular events were assesed in our study

Comment 3: The authors state that monocytes from the LTEC group are associated with an expansion of the intermediate monocyte subset. However, the data in Figure 1 show a wide range of levels across different donors within this group. It may be helpful to divide the LTEC group into subgroups and analyze potential contributing factors.

Response: we agree with the reviewer that there is a wide inter-individual variability for intermediate monocyte population in the LTEC group. See response to comment #2 for a potential explanation of this variability. Following the reviewer suggestion we have divided the LTEC group into two groups according to the level of intermediate monocytes (above and below the median value for the entire LTEC group) and have analyze potential differences between these two groups in terms of age, CD4 counts, length of infection and length of EC status. Interestingly we found that those LTEC with levels of intermediate monocyte population above the median had significantly higher length of EC status. Moreover, in the LTEC group, there was a significant correlation between years of EC status and level of intermediate monocyte population. This new information has been added into the revised version of the manuscript (discussion section, paragraph 9).

Comment 4: For Figure 2, the data presentation is unclear. Please add subfigure labels (e.g., Figure 2A, 2B, etc.) to each panel to help readers easily match the description with the figures. Additionally, in the lower part of the figure, you mention displaying data for the LTEC and OFFART groups, but each panel includes data from all four groups. Please clarify this inconsistency.

Response: we completely agree with the reviewer that the presentation of figure 2 is confusing. Figure legend does not precisely describe what is shown in the figure. Accordingly we have modified the figure labels and figure legend to more adequately describe the figure.

Comment 5: Overall, only performing a phenotypic study may not be sufficient to characterize monocyte perturbation in LTEC. I suggest conducting ex vivo studies to assess monocyte function, such as evaluating whether monocytes from LTEC exhibit increased antigen presentation or cytokine production etc...

Response: we agree with the reviewer that studies aimed to test the functionality of monocytes will add value to the phenotypic study that is presented in the manuscript, and could potentially support some of the hypothesis proposed in the discussion, such as for example a correlation between the level of intermediate monocyte subset and a better ability to present antigens or to produce certain cytokines upon stimulation. However due to cell sample availability we couldn’t perform any of these functional studies. This has been included as a limitation of the study in the revised version of the manuscript (discussion section, paragraph 10).

Round 2

Reviewer 3 Report

Comments and Suggestions for Authors

The authors have performed all the suggested experiments and addressed my concerns.

Author Response

Comment #1: The authors have performed all the suggested experiments and addressed my concerns.

Answer: We are very grateful for this feedback and the reviewer's work, which enabled us to improve the manuscript.

Reviewer 4 Report

Comments and Suggestions for Authors

The author has addressed most of my concerns, except for the functional test mentioned in my last comment.

Author Response

Comment #1: The author has addressed most of my concerns, except for the functional test mentioned in my last comment.

Response: as we mentioned in our previous response to the reviewers comment, we completely agree with the reviewer that testing the functionality of monocytes will add value to the phenotypic study that is presented in the manuscript,. However sample availability percluded us from performing this functional analysis. To better interpret the results of our study, this fact has been included as a limitation of the study in the revised version of the manuscript. Nonetheless, we feel that the findings of this phenotypic study are significant and advance our knowledge of the pathophysiology of HIV infection by demonstrating the presence of significant phenotypic changes in monocytes from HIV patients with long-term spontaneous control of viral replication (LTEC). This suggests the critical role of these cells in aspects of pathogenesis, such as the mechanisms governing disease control or the mechanisms causing the persistent inflammatory state.